# A Review on Surface Finishing Techniques for Difficult-to-Machine Ceramics by Non-Conventional Finishing Processes

**DOI:** 10.3390/ma15031227

**Published:** 2022-02-07

**Authors:** Lida Heng, Jeong Su Kim, Jun Hee Song, Sang Don Mun

**Affiliations:** 1Department of Mechanical Design Engineering, Jeonbuk National University, Jeonju 54896, Korea; henglida1@gmail.com; 2Department of Energy Storage/Conversion Engineering of Graduate School, Jeonbuk National University, Jeonju 54896, Korea; kjs1592@jbnu.ac.kr; 3Division of Convergence Technology Engineering, Jeonbuk National University, Jeonju 54896, Korea

**Keywords:** non-conventional finishing processes, advanced ceramics, difficult-to-machine ceramics, surface roughness, artificial hip joint

## Abstract

Ceramics are advanced engineering materials in which have been broadly used in numerous industries due to their superior mechanical and physical properties. For application, the industries require that the ceramic products have high-quality surface finishes, high dimensional accuracy, and clean surfaces to prevent and minimize thermal contact, adhesion, friction, and wear. Ceramics have been classified as difficult-to-machine materials owing to their high hardness, and brittleness. Thus, it is extremely difficult to process them with conventional finishing processes. In this review, trends in the development of non-conventional finishing processes for the surface finishing of difficult-to-machine ceramics are discussed and compared to better comprehend the key finishing capabilities and limitations of each process on improvements in terms of surface roughness. In addition, the future direction of non-conventional finishing processes is introduced. This review will be helpful to many researchers and academicians for carrying out additional research related to the surface finishing techniques of ceramics for applications in various fields.

## 1. Introduction

Ceramics are advanced engineering materials that have superior mechanical properties, high strength, high hardness, low thermal conductivity, and chemical inertness [1,2,3,4]. Additionally, they have superior biocompatibilities, high abrasion resistance, and corrosion resistance [5,6,7]. These advantageous mechanical and physical properties make them highly applicable to various high-tech industries, including medical technology, semiconductor, mechanical engineering, aerospace, automotive, ballistic armor, electronic, and cutting tools [8,9,10]. The most broadly used ceramics include zirconia (ZrO_2_), alumina (Al_2_O_3_), silicon carbide (SiC), and silicon nitride (Si_3_N_4_) [11,12,13,14,15]. Their mechanical properties are superior to those of metal materials and polymers. ZrO_2_ and Al_2_O_3_ are the most broadly used fine ceramics that are often utilized as inert material, which are widely employed in the medical industry (e.g., orthopedic surgery, artificial hip joints, unicondylar knee prostheses, ceramic crowns, implant systems, etc.), as reported by Heng et al. [16]. SiC is an artificial compound that is composed of commercial silica sand and carbon [17]. It supplies the best combination of heat resistance, low density, and light weight with low friability, and it maintains its strength at high temperatures up to 1500 °C [18,19]. Its superior mechanical properties make it ideal for utilization in engineering applications [20]. SiC has been chiefly utilized in cutting tools and grinding wheels for removing the metallic material from the surface of a workpiece with superior hardness and achieving high surface accuracy [21,22]. Si_3_N_4_ is a fine ceramic with superior mechanical properties such as superior corrosion resistance, high degree of toughness and thermal shock resistance at high temperatures, making it suitable for applying to engine parts (e.g., ball bearings, turbocharger rotors, cutting tools, etc.) [23,24]. The real and possible applications of ceramics are shown in Figure 1. The prescribed mechanical and physical properties of ceramics are shown in Table 1. Due to their high hardness, high young modulus, low density, and low thermal conductivity, these ceramics are applicable to many applications.

However, before they are employed in such applications, the products or components made using the ceramics must be manufactured with both high quality and efficiency with attributes such as high-quality surface finish, clean surfaces, dimensional conformity, and high accuracy of shape, in order to prevent and minimize thermal contact, adhesion, friction, and wear [16,39]. The ceramics are defined as difficult-to-machine materials that are present challenges to machining by the conventional finishing processes [40,41]. In addition, the inherent brittleness of ceramics poses further finishing problems [42]. Because of these difficulties, the conventional finishing processes face challenges meeting the specifications required of ceramics, including high-quality surface finish, dimensional conformity, and low tolerance.

The conventional finishing processes include processes such as grinding [43], turning [44], milling [45], lapping [46], super-finishing [47], and honing [48]. These conventional processes are limited in that it is difficult to control what happens during processing and in that they entail the application of high pressures, which can cause deep cracks in the finished surface [49]. As ceramics have limitations, such as poor tensile strength, low ductility, and inherent brittleness, the limited control and high pressure applied by these conventional processes can cause damage and cracks on the surface of the ceramics during the surface finishing process.

To overcome these problems, many researchers have adopted various non-conventional finishing processes (i.e., magnetic abrasive finishing (MAF), magnetorheological finishing (MRF), and clustered magnetorheological finishing (CMF)) to achieve smooth surface of advanced ceramics. The MAF process is a non-conventional finishing technique that was invented to produce a highly smooth surface and high workpiece form ultra-precision on round bar and inner surface materials using magnetic fields to govern a flexible magnetic abrasive brush during the finishing process [50,51,52]. The MAF process was utilized to achieve a high-quality surface finish for ceramic materials such as an Al_2_O_3_ fine ceramic round bar (Ø 3 mm × 60 mm) [39], a ZrO_2_ ceramic cylindrical bar (Ø 0.8 mm × 50 mm) [53], an Al_2_O_3_ ceramic plate (100 mm × 100 mm × 2.5 mm) [54], and an Al_2_O_3_ ceramic tube (Ø 50 mm × 100 mm) [55]. The magnetorheological finishing (MRF) process is another non-conventional finishing technique that has been used to finish components without subsurface damage [56]. The MRF process has been successfully utilized for achieving the high-quality surface finish of materials such as various optical glasses [57,58,59] and ceramics [60]. Hu et al. [61] proposed a form of MRF process based on permanent magnetic yoke excitation for achieving a high-quality surface finish on flat zirconia ceramics (40 mm × 40 mm × 1.1 mm) using MR fluid, which is the mixture of CI particles, abrasives, an additive agent and deionized water. Clustered magnetorheological finishing (CMRF) is a novel finishing process which can be used to achieve a fine surface finish on various materials, including silicon nitride ceramics [62], glass [63], and crystal silicon substrates [64]. The CMRF process is feasible for reducing the surface roughness to the nanometer range (2.69–9 nm) [64].

In this review paper, the finishing principles, finishing characteristics, current limitations, capabilities, and influence of these non-conventional finishing processes are discussed in detail and compared with regard to their ability to enhance the surface roughness and surface topography of difficult-to-machine ceramics. Finally, the future development of a new finishing procedure using a multi-axis/multi-faceted finishing technique for the surface finishing of materials of complex shapes, such as artificial hip joint ceramic components, is introduced.

## 2. Non-Conventional Finishing Processes

In recent years, there has been active development of novel non-conventional finishing processes and improvement of conventional processes to achieve the high-quality surface finish of difficult-to-machine materials such as ceramics, titanium alloys, tungsten, cobalt, and inconel 718 [65,66,67,68,69]. Generally, the characteristics of difficult-to-machine materials include high hardness, high strength, high brittleness, heat generation, low thermal conductivity, and poor surface quality [70,71,72,73]. The limitations of conventional processes are that they use high pressures with limited control during the finishing process [74]. Therefore, the workpiece surface to be finished can be damaged, or small cracks can form on the final surface after finishing. In addition, a high temperature can be generated on the surface of a workpiece with these conventional processes, for instance, through grinding, lapping, and horning. Moreover, these conventional processes have certain limitations in the finishing of materials with complicated shapes, require long processing times, and are not a cost-effective option for finishing small precision devices [49].

Unlike conventional processes, the non-conventional finishing processes have major advantages in that they use the extremely low normal force with small cuts produced by micro abrasive tools, which enables the production of a superior surface finish and a damage-free surface [75]. Additionally, these processes can also be applied to the finish of both large and micro-scale materials [16,76,77]. These non-conventional finishing processes, including cylindrical MAF, plane MAF, internal MAF, MRF, and CMRF, are described in the following sections.

### 2.1. Cylindrical Magnetic Abrasive Finishing (MAF)

A cylindrical MAF is a non-conventional finishing process which is used to produce a high-level surface quality on the object with a mirror surface level [51,78,79]. Chang et al. [22] proposed a cylindrical MAF for the external finishing of SKD11 material (Ø 15 mm × 100 mm) using unbonded magnetic abrasives. This process was also used by Heng et al. [80] for the external finishing of cylindrical magnesium alloy bars (Ø 3 mm × 50 mm) using a combination of carbon nanotube (CNT) particles and magnetic abrasive tools. His results indicated that the surface roughness of a magnesium alloy bar was improved to 0.02 μm from its original *Ra* value of 0.21 μm after finishing by this process. Mun et al. [81] also achieved high external surface finishing of an STS 304 bar by a cylindrical MAF process. 

According to their study, the cylindrical MAF process is able to obtain smooth surface finishes on various cylindrical workpiece materials of various sizes. Due to the numerous potential advantages of this process, Heng et al. [16] chose it for the external surface finishing of a ZrO_2_ ceramic cylindrical bar via different magnetic pole designs such as a magnetic pole with a sharp edge, a 2-mm square edge, and a 5-mm round edge.

#### Principle and Function

In the cylindrical MAF process, a cylindrical workpiece is positioned in the middle of the N- and S-pole of magnetic poles. The finishing gap between the cylindrical workpiece surface and the magnetic pole is filled with a mixture of magnetic abrasive tools, which generally consists of Fe powder, abrasive particles, and a lubricant [82]. The magnetic abrasive particles are magnetically joined together between two magnetic poles along the magnetic force lines to form a flexible magnetic abrasive brush (FMAB). This FMAB constantly acts against the rotating or vibrating surface of the workpiece, resulting in the removal of unevenness from the surface of the workpiece. The setup for applying the cylindrical MAF process to the external surface finishing of a ZrO_2_ ceramic by Heng et al. [16] is shown in Figure 2. The procedure used in his study involves fastening the ZrO_2_ in a chuck and rotating it at 35,000 rpm inside the vibrating magnetic abrasive tools, a mixture of Fe powder (200 μm), diamond abrasives (1 μm), carbon nanotube (CNT) particles (0.04 μm), and light oil (0.2 mL). A ZrO_2_ ceramic (dimension: 0.8 mm × 50 mm) used in his study is shown in Figure 3a and a schematic diagram of the different magnetic pole shapes is shown in Figure 3b. According to his results, the surface roughness (*Ra*) of ZrO_2_ was significantly decreased from 0.18 μm to 0.02 μm by the cylindrical MAF process under optimal conditions (magnetic pole shape: 2-mm square edge, abrasive grain: 1-μm, magnetic pole vibration: 8-Hz, rotation speed: 35,000 rpm, and processing time: 40 s). AFM surface topography images of the ZrO_2_ ceramic are shown in Figure 4. Before the finishing process, the surface of the ZrO_2_ was uneven and had many peaks, with an average peak height of 0.18 μm (Figure 4a). After the finishing process under optimal conditions, the irregular peaks were entirely eliminated from the surface of the ZrO_2_, and the surface condition was much finer (Figure 4b).

### 2.2. Plane Magnetic Abrasive Finishing (MAF)

Plane MAF is a non-conventional finishing process, in which the magnetic abrasive brush is formed inside the finishing gap between the rotating magnetic pole and the surface of the reciprocating plate workpiece upon the application of a magnetic field [83,84]. Generally, the finishing action of plane MAF is generated by the magnetic field acting on the magnetic abrasive against the surface of plate workpiece, causing micro-chip removal, which gradually reduces the surface roughness value of the plate workpiece’s surface. Many researchers have investigated the finishing characteristics of this process, and they concluded that this process has the capability to achieve a smooth surface finishing of plate workpiece surfaces [85,86,87]. Its major advantage is that it successfully achieves a high-quality surface finish of numerous materials, including SUS304 [88,89], brass [90,91], AISI 1018 mild steel [92], 5052 aluminum alloy [93], AZ91 magnesium alloy [94], and glass [95]. Due to its major advantages, Zou et al. [54] studied the possibility of processing alumina ceramic plates via plane MAF by employing an alternating magnetic field.

#### Principle and Function

The schematic of a plane MAF setup for the surface finishing of alumina ceramic plates based on the work of Zou et al. [54] is shown in Figure 5. To generate the reciprocating and rotational motion of the electromagnet, the electromagnet was fastened to motor A, which enabled the electromagnet’s rotation with a rotational speed that can be adjusted by motor controller A. Both the electromagnet and motor were positioned on the magnet holder, and its reciprocating motion was controlled by motor B, which made them reciprocate in one direction along the X-axis. A mixture of magnetic abrasive particles (mostly consisting of iron powder and abrasive particles) filled the gap between the ceramic plate’s surface and the tray. During the finishing process, when the voltage was supplied to an electromagnet, a magnetic field was generated, causing the abrasive particles to magnetically attract each other. Thus, the particles were transformed into a magnetic cluster inside the gap between the surface of the alumina ceramic plate and the tray. The reciprocating and rotational motion of the magnetic pole causes the frictional action between the ceramic plate’s surface and the magnetic cluster, removing the peaks and micro-chips from the alumina ceramic plate’s surface. Using the plane MAF process, the surface roughness of the ceramic plate was decreased from 244.6 to 106.3 nm within 80 min of processing time, as reported by Zou et al. [54].

### 2.3. Internal Magnetic Abrasive Finishing (MAF)

The internal MAF process for the internal finishing of tube objects was first proposed in 1995 by Shinmura et al. [96]. As reported by many researchers, this process has the main advantage of finishing the internal surface of tubes with various materials and sizes [97,98]. Due to its major advantage, Yun et al. [55] has applied an internal MAF process that uses ultrasonic vibration for finishing the surface of alumina ceramic tubes.

#### Principle and Function

A schematic of the internal MAF process for the surface finishing of an alumina ceramic tube using the ultrasonic vibration L-shaped magnet is shown in Figure 6. The internal MAF process is a non-conventional finishing technique in which the magnetic abrasive particles are inserted inside the internal surface of a tube workpiece. These particles are controlled by magnetic forces which are generated by the N-pole and S-pole of magnets [99,100]. A front view of the internal MAF process for the surface finishing of an alumina ceramic tube is shown in Figure 7. During the processing, the abrasive particles are bound in the form of flexible magnetic abrasive brush that presses against the rotating internal surface of the workpiece, reducing the surface roughness (*Ra*) of the internal surface. To improve the high finishing efficiency for an alumina ceramic tube in the internal MAF process, the use of an ultrasonic vibration system was explored by Yun et al. [55]. 

The processing procedure used in this study employs an L-shaped magnet and magnetic abrasive tools (mixture of iron powder and diamond abrasives) positioned inside the ceramic tube. Due to the strong magnetic field, the magnetic abrasive tools formed a flexible magnetic abrasive brush for acting against the rotating internal surface of the ceramic tube. During the finishing process, when the ceramic tube was rotated, the outside magnet was reciprocally moved along the ceramic tube axis, which drove movement of the internal magnet within the magnetic abrasive brush along the ceramic tube axis. At the same time, an L-shape magnet with a magnetic abrasive brush started to vibrate with a certain frequency via adherence to ultrasonic vibration. Using this procedure, the magnetic abrasive brush can effectively remove the unevenness from the internal surface of ceramic tube. Yun’s results showed that the Ra of an alumina ceramic tube was reduced from 1.1 µm to 0.03 µm by internal MAF accompanied with ultrasonic vibration under optimal conditions [55].

### 2.4. Magnetorheological Finishing (MRF)

The MRF process is a non-conventional micro finishing technique which has been used to finish components without causing subsurface damage. In this process, a workpiece surface is finished in a magnetorheological finishing fluid which is composed of carbonyl iron power and magnetic abrasives suspended in a carrier liquid [101,102]. In this process, the magnetorheological fluid is delivered into a rotational wheel, and pulled against the rotating wheel surface using a magnetic field, then the workpiece is plunged into the ribbon of MR fluid, which enables the achievement of a high-quality surface finish [103]. Generally, the MRF process is widely applied for mirror finishing a wide variety of highly brittle materials (e.g., glasses and hard crystals) [104,105]. MRF uses a very low normal force, with micro cuts generated by magnetic abrasives, which can achieve a high-quality surface finish without surface damage. Due to these advantages, Hu et al. [61] used the MRF process to obtain a smooth surface on a ZrO_2_ ceramic plate. As ZrO_2_ is a difficult-to-machine material because of its high brittleness and high hardness, it could reduce the finishing efficiency of the general MRF process. To combat this potential problem, a permanent magnetic (PM) yoke excitation was added to the MRF process and the feasibility of this setup was examined [61].

#### Principle and Function

The schematic and experimental setup for the MRF process of ZrO_2_ surfaces are shown in Figure 8 and Figure 9 [61]. The experimental setup of this process consists of a lapping machining, a speed governor, a trough, a magnetic yoke, a workpiece fixture, and a ZrO_2_ workpiece. A ZrO_2_ workpiece is installed at a fixture that enables rotation along the B-axis. The finishing trough is fastened on the A-axis, which enables its rotation opposite to the B-axis. The PM yoke was installed below the ZrO_2_ workpiece, and it was used to enlarge the finishing area between the ZrO_2_ surface and the MR ribbon. During the finishing process, when the magnetorheological (MR) fluid passed over the magnetic field gradient, it stiffened to form a rectangular magnetorheological fluid ribbon within one millisecond. Due to the magnetic field gradient, the magnetic finishing abrasives float upward and congregate at the interface of the ZrO_2_ and the magnetorheological fluid ribbon. By the pressure of magnetorheological ribbon, the finishing abrasive acts strongly on the ZrO_2_ surface, resulting in high quality surface finishing. After the finishing process, the ZrO_2_ had a mirror surface with surface roughness values less than 1 nm. The evolution of the surface topography of a ZrO_2_ plate from before to after MRF is shown in Figure 10. Before finishing, extremely rough peak-valley structures of micrometer sizes can be seen throughout the surface of ZrO_2_, corresponding to a surface roughness of 71.976 nm (*Ra*) (see Figure 10a). After MRF, it was found that all peak-valleys were entirely removed and the surface condition was much smoother than before finishing. The final surface roughness value was 0.702 nm (*Ra*) after finishing with a diamond abrasive within 30 min (see Figure 10b). Figure 10c shows images of ZrO_2_ surface conditions before and after finishing via the MRF process. It can be seen that the text on the bottom was not reflected by the surface of ZrO_2_ before finishing due to poor ZrO_2_ reflectivity (Figure 10c, left). After MRF, the reflection of the text was clearly projected by the ZrO_2_ surface, as the final surface finishing increased the smoothness of the ZrO_2_ surface, thereby improving reflectivity.

### 2.5. Clustered Magnetorheological Finishing (CMRF)

CMRF is a novel finishing process in which dynamic magnetic fields are formed via the synchronous eccentric rotation of numerous magnets [64]. This process has been generally used for obtaining super smooth surface finishes of spherical materials [62,64,106]. Xiao et al. [62] proposed a novel CMRF method for the surface finishing of the silicon nitride (Si_3_N_4_) ceramic balls.

#### Principle and Function

The CMRF process for Si_3_N_4_ balls is shown in Figure 11. The CMRF setup is composed of upper and lower finishing heads, the Si_3_N_4_ balls, and an MRF slurry (mostly a mixture of abrasive particles, Fe powder, and grinding fluid). In the CMRF process, the ceramic balls are placed inside a groove of the lower finishing head, and they are surrounded by MRF slurry. The MRF slurry forms a magnetic chain shape because of the cluster magnetorheological effect caused by the N-pole and S-pole of the magnets [62]. During the CMRF process, both the lower and upper finishing heads are rotated around their respective spindles. The spindle of the upper finishing head simultaneously revolves around the spindle of the lower finishing head. Unevenness or micro-cracks on the surfaces of the Si_3_N_4_ balls are eliminated by the abrasion between the spherical surfaces of the Si_3_N_4_ balls and the MRF abrasives. According to the results of Xiao et al. [62], after the CMRF process, the surface quality of Si_3_N_4_ balls is greatly improved, with the surface roughness value decreasing from 63 nm (*Ra*) to 4.35 nm (*Ra*) under optimal CMRF conditions.

SEM surface topography and optical images of Si_3_N_4_ balls before and after finishing by CMRF process are compared in Figure 12. Before the CMRF process, dimples, craters, and cracks can be found everywhere on the surface of the Si_3_N_4_ balls (see Figure 12a). In contrast, after CMRF, most of the dimples, craters, and cracks were entirely removed from the surface of the Si_3_N_4_ balls (see Figure 12b). A change in the surface quality can also be observed in optical images of the Si_3_N_4_ balls. Before the CMRF process (Figure 12c), the reflectivity of balls is limited, so that the reflection of nearby text is not clearly seen on the surface of the Si_3_N_4_ balls. After CMRF, the reflectivity of the Si_3_N_4_ balls is vastly improved, clearly reflecting nearby text (Figure 12d).

## 3. Summary and Analysis of Non-Conventional Finishing Processes

### 3.1. Comparison between Each Finishing Processes

From these studies, it is evident that all the discussed processes can successfully achieve the high-quality surface finish of difficult-to-machine ceramics with low values of surface roughness (*Ra*). To improve the finishing efficiency, researchers have applied critical parameters to their finishing processes, which have had major influence on improvement in surface roughness (*Ra*). In this study, a percentage of improvement in surface roughness equation was implemented in order to compare the finishing capabilities of each process on the improvements in surface roughness. Equation (1) expresses the percentage of improvement in the surface roughness (*Ra*) as a function of the non-conventional finishing processes. *BFP* is the value of (*Ra*) before the finishing process, *AFP* is the value of (*Ra*) after the finishing process, and *PISR* is the percentage improvement in surface roughness (*Ra*). According to Equation (1), it was found that the surface roughness improvement for each of these non-conventional finishing processes was greater than 56%. When cylindrical MAF, plane MAF, internal MAF, MRF, and CMRF process were used for finishing the surface of ceramics, they showed improvements of 88.88%, 56.54%, 97.27%, 99.02%, and 93.09%, respectively.

Table 2 indicates the values of surface roughness (*Ra*) obtained by these different non-conventional finishing processes. These data suggest that for achieving high surface quality of ceramic material bars, the cylindrical MAF process can be used because it can produce the best results within the short finishing time of only 40 s. A smooth surface on a ceramic plate can be achieved by different processes such as plane MAF and MRF process, with the best results found when the MRF process was used. The MRF process required only 30 min to obtain the optimal result, and the improvement in surface roughness by this process was 99.02%. In contrast, the plane MAF process required 80 min to obtain the optimal result, and the PISR of this process was only 56.54%. Therefore, it was found that MRF was best suited to finish the surface of plane ceramic material. As seen in Table 2, the CMRF process successfully achieves smooth surfaces on ceramic balls, reducing surface roughness values from 63 nm to 4.35 nm within 60 min of finishing time. The internal MAF process successfully achieves smooth surfaces in ceramic tubes, reducing the surface roughness values from 1.1 to 0.03 µm within 50 min of finishing time. This can confirm that non-conventional finishing processes have the capability to achieve the high-quality surface finish of difficult-to-machine ceramics. Moreover, the parameters affecting the material topography or cracking on the surface of ceramics during and after the experiments are not shown in these studies. According to these studies, the broad applicability of these non-conventional finishing processes can be found to be as described below.

The plane MAF and MRF process can be used in the optical industry for achieving the high-quality surface finish of transparent ceramic lenses, and these processes can be used in the semiconductor industry for achieving the high-quality surface finish of the electrostatic ceramic chuck component. The cylindrical MAF process can be used in the electronics industry for achieving the high-quality surface finish of a ceramic needle pin, and in the medical industry for items such as ceramic tooth screws. The internal MAF process can be used in the automotive industry for achieving the high-quality surface finish of ceramic nozzles, and for ceramic heat pipes used in heat transfer applications. With the CMRF process, it is possible to achieve high-quality surface finish of the ceramic bearing ball used in electric motors and in the aerospace applications.

However, with these processes it is difficult to achieve the high-quality surface finish of the products with complex shapes which are currently used in the medical applications (i.e., ceramic femorals, ceramic liners, knee prosthesis, etc.), and in gas turbine engine applications such as ceramic turbine rotors. This is due to the limitations of their experimental device that make it difficult to process complexly shaped surface ceramics.

Therefore, in order to achieve the high-quality surface finish of the ceramic component with complex shapes, new developments and a new finishing procedure related to these non-conventional finishing processes are required.
(1)$$PISR=\frac{BFP-AFP}{BFP}\times 100\%$$

### 3.2. Processes Limitations

These non-conventional finishing processes have been shown to successfully finish the ceramic components of various shapes, including cylindrical bars, plates, tubes, and round balls. Despite the advantages of these processes, some limitations associated with these processes still remain.

The following limitations of these processes are listed below:-The plane MAF, internal MAF, MRF, and CMRF processes are very slow because they require large finishing time for difficult-to-machine materials such as ceramics.-Products with freeform shapes, including elliptical, conical, wrinkle, and concave shapes are impossible to finish via these processes due to the variations in the finishing gap between the tools and the workpiece. Some limitations associated with recent non-conventional finishing processes are shown in Table 3.

## 4. Future Directions

Ceramics are advanced materials that are widely used in many high-tech industries. Ceramic components are highly priced when compared to other advanced materials such as cobalt chromium (Co-Cr) alloys, titanium alloys, and stainless steel. The challenges related to the surface finishing of ceramics include long finishing times, high energy consumption, high rate of tool wear, and poor surface finishes by conventional finishing processes. In spite of these difficulties, non-conventional finishing processes have been shown to successfully achieve the high-quality surface finish of advanced ceramics. However, most of these processes have major problems with finishing times, as they require long finishing times for hard materials such as advanced ceramics. In addition, these processes are not applicable for freeform surfaces.

To overcome these problems, novel advanced research regarding finishing processes must be pursued. Novel advanced processes must go beyond the existing capabilities of the non-conventional finishing processes. To this end, future work could apply a multi-axis/multi-faceted surface finishing technique to the cylindrical MAF process for finishing ceramic components with spherical shapes, such as an artificial hip joint (Figure 13). The artificial hip joints used in medical applications is shown in Figure 13: (a) ceramic spherical ball joint [109], (b) metal spherical ball joint [109], and (c) schematic view of spherical ball joint. The CMRF process can finish ceramic workpieces with spherical shapes. However, this process requires lengthy finishing times to achieve high quality surface finishes on workpieces.

Unlike the CMRF process, the rotational motions are applied in the X-, Y-, and Z-directions in the cylindrical MAF process. Rotational motion can be added to the spherical ball workpiece using a step motor, enabling workpiece rotation up to 10,000 rpm in the Y-direction. The rotational motions of both magnetic pole systems are generated by step motors, so they can simultaneously rotate in the X- and Z-directions. The addition of high rotational speed to the spherical ball workpiece can increase the relative action between the surface of spherical ball and the magnetic abrasive tools, resulted in the achievement of a high-quality surface finish on spherical ball workpiece within a short finishing time less than 10 min. The working principle of a cylindrical MAF process using a multi-axis/multi-faceted finishing technique for the surface finishing of ceramic components with spherical shapes such as artificial hip joints is shown in Figure 14. The spherical ball workpiece is inserted into the particulate magnetic abrasive brush of the N-pole and S-pole of the magnetic pole (see Figure 14a). To generate the rotational motion of the workpiece, the workpiece is fastened to a step motor. Two sets of magnetic pole parts are attached to the permanent magnets, and each is fastened to the step motor using a magnet fixture. During the finishing process, the magnetic abrasive tools (a mixture of diamond abrasive particles, iron powder, and lubricant) are placed inside the gap of workpiece and magnetic pole edges. Due to the strong magnetic forces from both the N- and S-poles of the magnets, these magnetic abrasive tools are attracted to each other along the magnetic force lines. These magnetic abrasive tools form a flexible magnetic abrasive brush, acting strongly against the surface of the spherical ball workpiece. When rotational motion is applied to the X-, Y-, and Z-directions, micro removal actions by the magnetic abrasive tools on the workpiece surface are generated. Via this procedure, the spherical ball workpiece can be finished with a superior smooth surface and high form precision in roundness. As the high rotational speed and diamond abrasive tools are applied in this process, the whole finishing process will take less than 10 min to obtain the optimal result.

## 5. Concluding Remarks

According to the above papers, the conclusions of this study are as follows:All the non-conventional finishing processes discussed in this review have been successful in achieving the high-quality surface finish of various difficult-to-machine ceramics with low values of surface roughness (*Ra*). When the cylindrical MAF, plane MAF, internal MAF, MRF, and CMRF process were used for finishing the surface of ceramics, they showed surface roughness (*Ra*) improvements of 88.88%, 56.54%, 97.27%, 99.02%, and 93.09%, respectively.Cylindrical MAF is used for finishing the surface of ceramic products with cylindrical shapes, internal MAF is used for finishing the internal surfaces of round tubes, CMRF is used for finishing the surface of spherical ceramics, and plane MAF and MRF are used for finishing the surface of planar ceramics. Between plane MAF and MRF, MRF can be assumed to be the better method for finishing the surface of plane ceramics. MRF requires shorter finishing times and achieves greater finishing efficiency when compared to plane MAF.Despite the significant advantages of these advanced finishing processes, some limitations still remain, such as (1) inapplicability to finishing freeform surfaces, (2) lengthy finishing times, and (3) difficulty finishing the surface of ferromagnetic materials.In the future, a multi-axis/multi-faceted surface finishing technique can be added to the cylindrical MAF process for the surface finishing of difficult-to-machine ceramic components used in medical applications such as artificial hip joints. This process would have major advantages such as short processing times and the ability to process all common materials, including ceramics, metals, and polymers.

## Figures and Tables

**Figure 1 materials-15-01227-f001:**
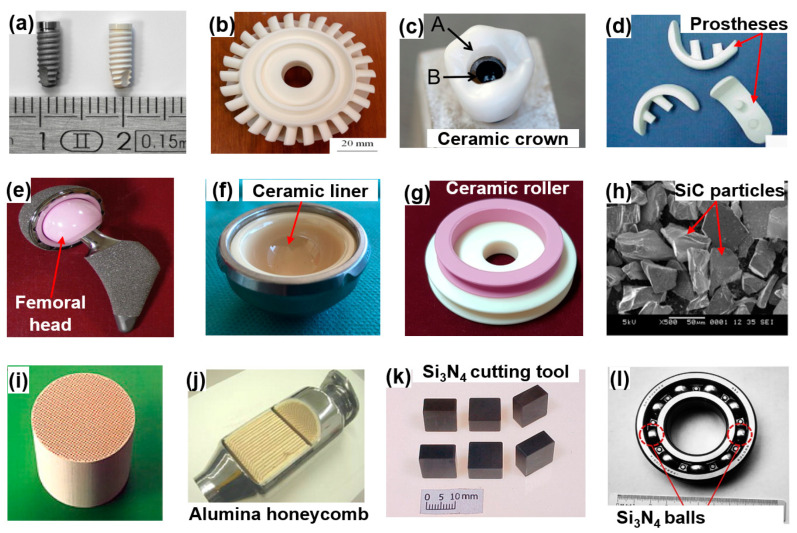
Real/possible applications of ceramics: (**a**) ceramic implant system in medical application [25], (**b**) ceramic turbine rotor [26], (**c**) ceramic crown [27], (**d**) unicondylar knee prostheses [28], (**e**) ceramic femoral head [29], (**f**) ceramic liner [30], (**g**) ceramic textile roller, (**h**) SEM image of SiC abrasive particles [31], (**i**) catalyst honeycombs used in automobile applications [32], (**j**) cross-section of a honeycomb [33], (**k**) Si_3_N_4_-based cutting tools [34], (**l**) Si_3_N_4_ ball bearings [35]. Reused with permission from Elsevier.

**Figure 2 materials-15-01227-f002:**
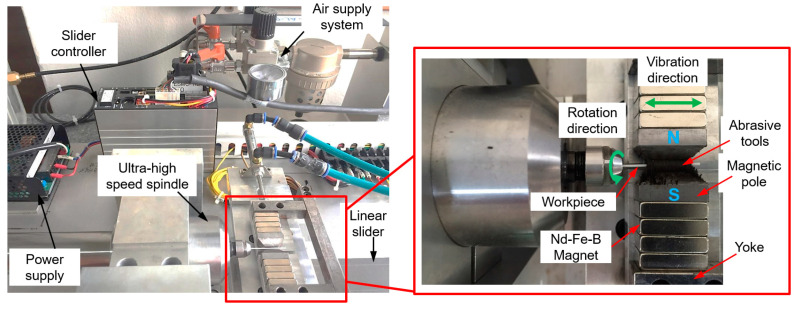
Cylindrical magnetic abrasive finishing experimental set up for ZrO_2_ ceramics, reused with permission from Elsevier [16].

**Figure 3 materials-15-01227-f003:**
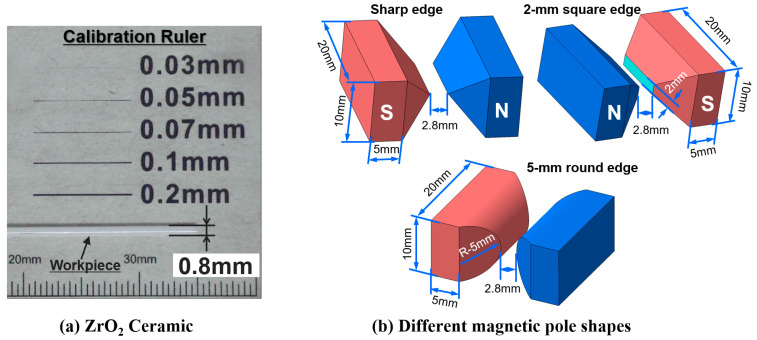
(**a**) ZrO_2_ ceramic workpiece (0.8 mm × 50 mm), (**b**) different magnetic pole shapes, reused with permission from Elsevier [16].

**Figure 4 materials-15-01227-f004:**
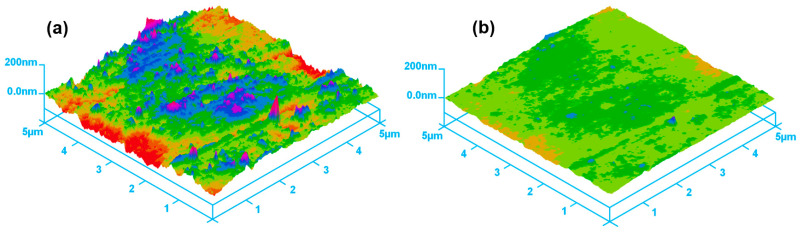
AFM surface topography images of ZrO_2_ ceramic: (**a**) before processing (*Ra*: 0.18 μm), and (**b**) after processing (*Ra*: 0.02 μm), reused with permission from Elsevier [16].

**Figure 5 materials-15-01227-f005:**
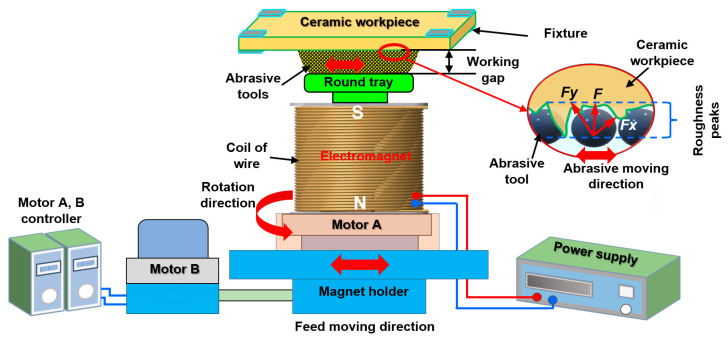
Schematic of a plane magnetic abrasive finishing setup for a ceramic plate workpiece, redrawn from Zou et al. [54].

**Figure 6 materials-15-01227-f006:**
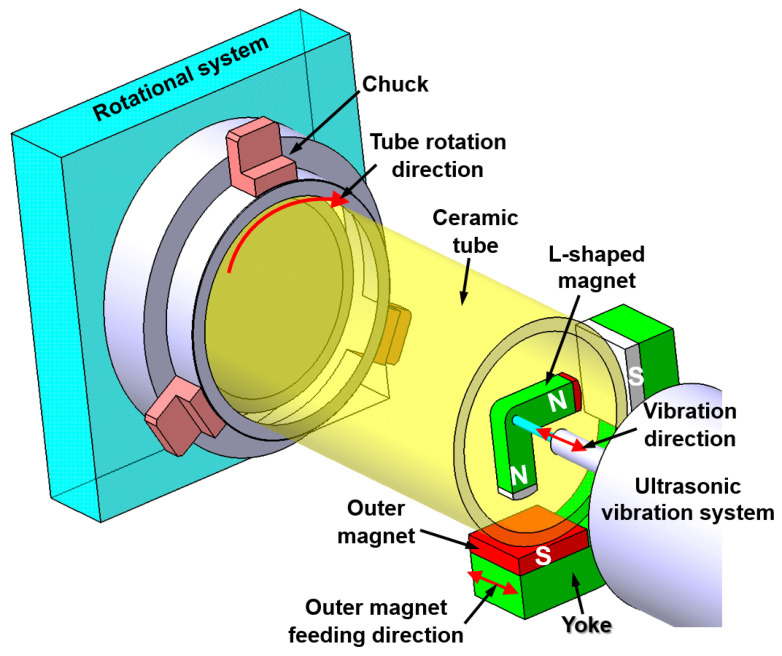
Schematic of the internal MAF process for the surface finishing of alumina ceramic tube using an ultrasonic vibration L-shaped magnet, redrawn from Yun et al. [55].

**Figure 7 materials-15-01227-f007:**
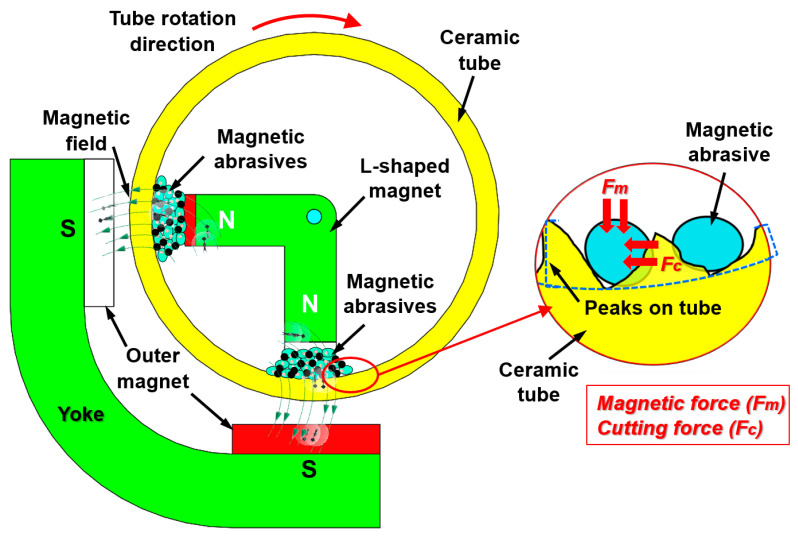
Front view of internal MAF process for the surface finishing of alumina ceramic tube, redrawn from Yun et al. [55].

**Figure 8 materials-15-01227-f008:**
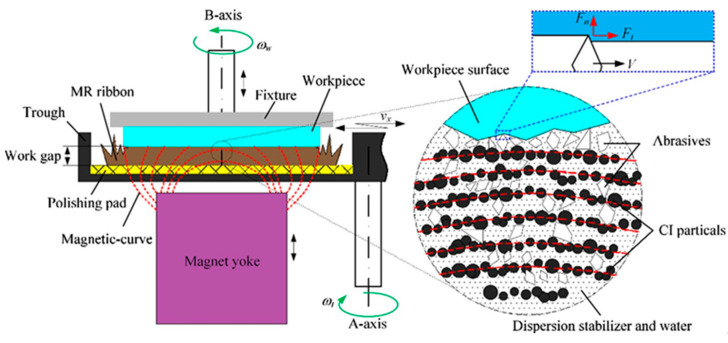
Schematic of magnetorheological finishing of ZrO_2_ ceramics, reused with permission from Elsevier [61].

**Figure 9 materials-15-01227-f009:**
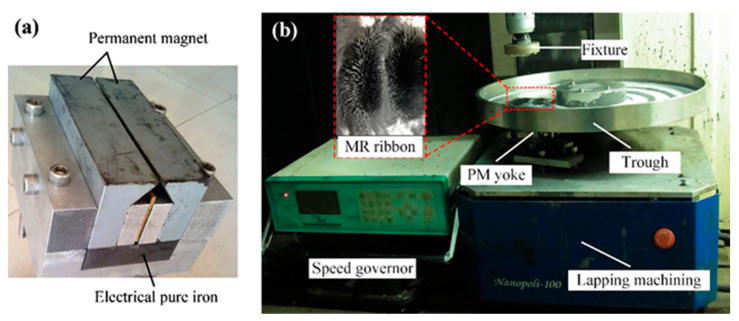
Experimental setup of magnetorheological finishing of zirconia ceramics **(a)** permanent magnetic yoke and **(b)** experimental setup, reused with permission from Elsevier [61].

**Figure 10 materials-15-01227-f010:**
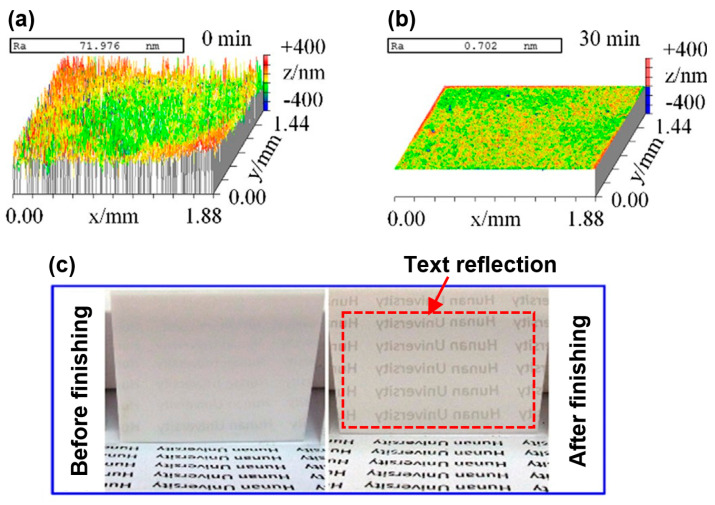
Surface conditions of ceramic plate: (**a**) surface topography of unfinished surface, (**b**) surface topography of finished surface, (**c**) images comparison between unfinished surface and finish surface, reused with permission from Elsevier [61].

**Figure 11 materials-15-01227-f011:**
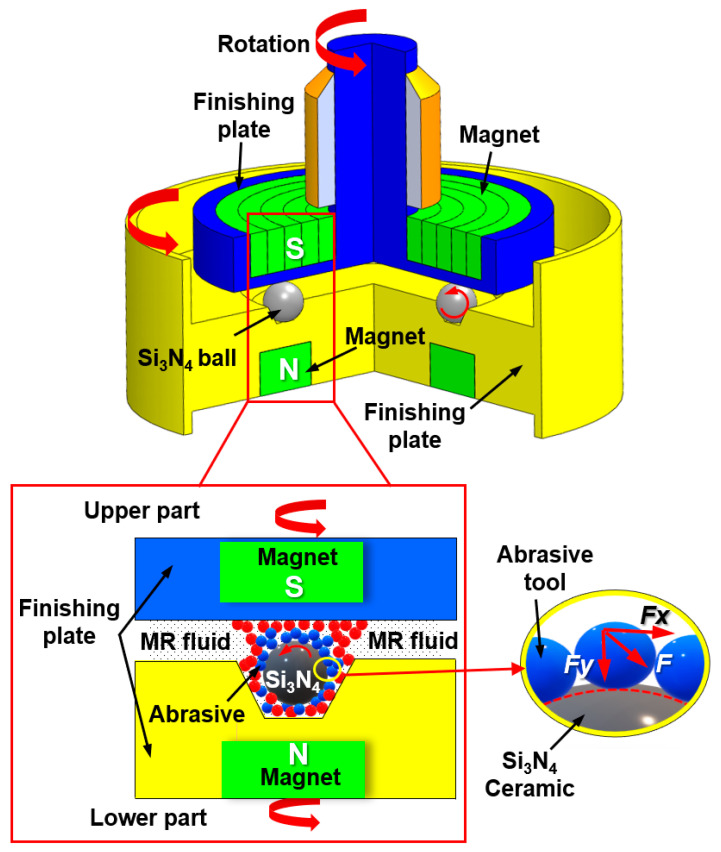
Schematic of the CMRF experiment for the surface finishing of Si_3_N_4_ balls, redrawn from [62].

**Figure 12 materials-15-01227-f012:**
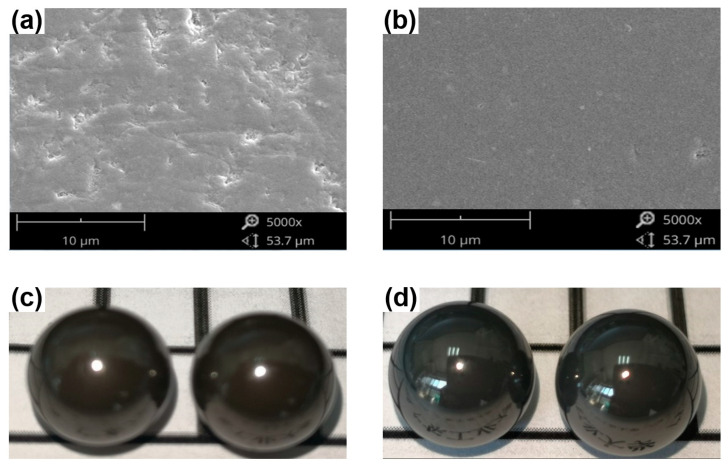
Surface conditions of Si_3_N_4_ ceramic balls: (**a**) SEM topography of unfinished surface, (**b**) SEM topography of finished surface, (**c**) optical image of unfinished surface, (**d**) optical image of finished surface. Data obtained from Ref. [62], under an open access CC BY 4.0 license.

**Figure 13 materials-15-01227-f013:**
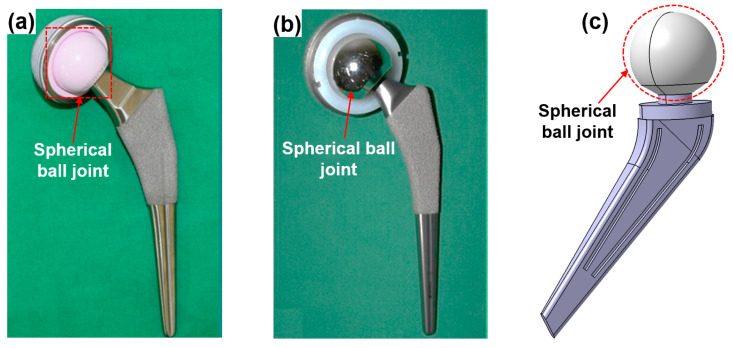
Artificial hip joint used in medical applications: (**a**) ceramic spherical ball joint, (**b**) metal spherical ball joint, and (**c**) schematic view of spherical ball joint. Data obtained from Hu et al. [109]. CC BY license.

**Figure 14 materials-15-01227-f014:**
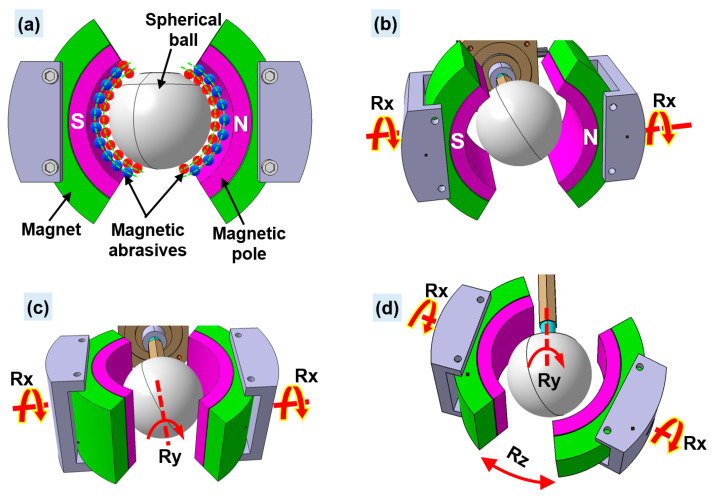
Working principle of a cylindrical MAF process using a multi-axis/multi-faceted finishing technique for the surface finishing of a spherical ceramic ball (**a**) spherical ball inside the magnetic abrasive brush (**b**) magnetic pole rotating on X-direction (**c**) spherical ball rotating on Y-direction and (**d**) magnetic pole rotating on Z-direction.

**Table 1 materials-15-01227-t001:** Prescribed mechanical and physical properties of ceramics [36,37,38]. Reused with permission from Elsevier.

Materials	Density (g/cm^3^)	Hardness (HRA)	Young Modulus (GPa)	Coefficient of Thermal Expansion (10^−6^/K)	Thermal Conductivity (J/cm.s.K)
Si_3_N_4_	3.25–3.35	92–94	304–330	3.2–3.5	0.155–0.293
Al_2_O_3_	3.6	91	-	-	0.25
ZrO_2_	5.6	88	200	8	1.8
SiC	3.16–3.2	-	410	4.4	1.2–1.8

**Table 2 materials-15-01227-t002:** Surface finish of advanced ceramics obtainable by non-conventional finishing processes.

No.	Non-Conventional Finishing Processes	Workpiece	Surface Roughness, (*Ra*)	Finishing Time	Percentage Improvement in (*Ra*), *PISR*
1	Cylindrical MAF	ZrO_2_ cylindrical bar	From 0.18 to 0.02 µm	40 s	88.88%
2	Plane MAF	Al_2_O_3_ ceramic plate	From 244.6 to 106.3 nm	80 min	56.54%
3	Internal MAF	Al_2_O_3_ ceramic tube	From 1.1 to 0.03 µm	50 min	97.27%
4	MRF	ZrO_2_ ceramic plate	From 71.976 to 0.702 nm	30 min	99.02%
5	CMRF	Si_3_N_4_ ceramic ball	From 63 to 4.35 nm	60 min	93.09%

**Table 3 materials-15-01227-t003:** Limitations of non-conventional finishing processes [49,64,107,108].

No.	Advanced Finishing Processes	Limitations
1	Cylindrical MAF	Not applicable to freeform surfaces. Difficulty regarding ferromagnetic materials.
2	Plane MAF	Not applicable to freeform surfaces. Requires lengthy finishing times for hard material.
3	Internal MAF	Not applicable to freeform surfaces. Requires lengthy finishing times for hard material.
4	MRF	Not applicable to freeform surfaces. Requires lengthy finishing times for hard material. Only applicable to optical materials and ceramics.
5	CMRF	Can only finish spherical surfaces. Requires lengthy finishing times for hard material.

## Data Availability

The data presented in this study are available upon request from the corresponding author.

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
