# Peer review of "A Review on Surface Finishing Techniques for Difficult-to-Machine Ceramics by Non-Conventional Finishing Processes"

_materials, 2022, doi:10.3390/ma15031227_

Round 1

Reviewer 1 Report

In the manuscript the finishing processes are well discussed highlighting the main characteristics, principles and limitations.

In particular, the main processes applied to ceramic materials for improving the surface finishing are introduced describing the processes’ characteristics through the presentation of several application cases.

The most interesting part is the section 4 which furnish a direct comparison between the finishing processes and their typical applications in terms of materials and range of surface roughness achievable.

The structure of the work is very clear and the description cover the main aspects related to each process.

Just 2 suggestion:

  1. some acronyms are not spelled out the first time they appear (e.g. CNT in section 3.1). please check the manuscript and clarify their meaning.
  2. The most innovative part of this review is represented by the section 4. Since this part reports very interesting aspects of the investigated processes I suggest to the author to implement this section for strengthening the manuscript, giving to the reader a complete overview of this comparison (both in terms of applicability and limitation).

Author Response

Dear Editor and Reviewers,

We would like to thank the editor and reviewers for careful and thorough reading of this manuscript and for the thoughtful comments and constructive suggestions, which help to improve the quality of this manuscript. The manuscript has been revised according to your comments. The attached file is already uploaded.

Best regards,

From: Prof. Sang Don Mun

Jeonbuk National University

Reviewer 2 Report

The authors undertook to prepare a review scientific communication on the subject of “ultra-precision finishing for difficult-to-machine ceramics by advanced finishing processes”. Unfortunately, I have already noticed a number of errors when reading the abstract and the introduction. Perhaps the most prominent of these is that the abstract and the introductions (especially the first and last paragraphs of the introduction) are largely the same. There's just been a rewording in many cases. These reruns can also be found in the rest of the thesis.

The authors copied all the diagrams of the thesis from other authors. This would not be a problem in itself, but overall the thesis does not give any additional results compared to the approximately 115 articles cited. In this form, the article is nothing more than an aggregation of all these 115 articles. A review article should also include the extra results, extra knowledge that separates from a simple summary. Since I have read many other review articles throughout my career, unfortunately, I do not consider the work satisfactory for publication in the journal "Materials".

Author Response

(The authors gave the same response as above.)

Reviewer 3 Report

Dear Sir/Madam   The manuscript is based on a comprehensive review related to the mechanical and physical properties of materials for different industrial applications. Different advanced finishing processes which could enhance the performance of these materials are also discussed in detail. In addition, the influence of different parameters and current limitations mentioned in the manuscript will provide a guide for the researchers of the field working in this area. In my opinion, the manuscript is well-written and will attract readers of this area. Therefore, I recommend it for publication in Materials.

Author Response

Dear Editor and Reviewers,

We would like to thank the editor and reviewers for careful and thorough reading of this manuscript and for the thoughtful comments and constructive suggestions, which help to improve the quality of this manuscript. Thank you very much for your kind words and the positive comments. We greatly appreciate your comments.

Best regards,

From: Prof. Sang Don Mun

Jeonbuk National University 

Reviewer 4 Report

In this paper, the advanced finishing processes with regard to ceramics were summarized, and numerous literatures regarding the magnetic abrasive finishing. I believe the review paper will be of great interest to your readership. However, some issues should be address before publication, my major comments are as follows:

  1. The title mentions the advanced finishing processes, but the primary literatures are related with magnetic abrasive finishing. Is the magnetic abrasive finishing equal to the advanced finishing processes? If not, please make the title more specific.
  2. The abstract clarifies an excess of research background, the related content should be more concise.
  3. Maybe it is more appropriate to merge the second section “Ceramic Applications” into the first section “Introduction”. The second section is incompatible with the subject.
  4. How is the author's review paper different from other review papers in this field, such as reference 23 and reference 24.
  5. In the third section, the authors introduce the magnetic abrasive finishing, which can achieve cylindrical, plane, internal, magnetorheological, and clustered magnetorheological processes. The principles and functions are introduced in the paper. Moreover, it also is required to weigh and analyze the development status and prospects of the field from his own perspective. When I read the review, I look forward to not only reading the references but also check if the author has some comments aim at references. It would be great if the review could enlighten me. Nonetheless, it's just a pure list of literatures.
  6. The authors should exhibit the magnetic abrasive finishing with respect to ceramics from the initial researches to the latest researches.

Author Response

(The authors gave the same response as above.)

Reviewer 5 Report

Dear Authors,

I have carefully reviewed your manuscript entitled “A Review on Ultra-Precision Finishing for Difficult-to-Machine Ceramics by Advanced Finishing Processes”, submitted for publication in the Materials journal and I would like to say that this manuscript can be published in the Materials journal in present form.

The authors discussed an important task of ultra-precision finishing of advanced ceramics, which are more and more attractive engineering materials. This well-known problem is a goal of many research projects. In this manuscript, the authors collected and discussed the research results presented by other authors in 115 articles. In general, the objectives of the review paper are clearly defined. The introduction provides a good, generalized background of the topic. The collected results are clearly explained and are presented in an appropriate format. The figures and tables show essential data; some of the data are also summarized in the text. The cited literature is relevant to the study and balanced. I cannot see any significant drawbacks.

Author Response

Dear Editor and Reviewers,

We would like to thank the editor and reviewers for careful and thorough reading of this manuscript and for the thoughtful comments and constructive suggestions, which help to improve the quality of this manuscript. Thank you very much for your effort and for your kind words with the positive comments. We greatly appreciate your comments.

Best regards,

From: Prof. Sang Don Mun

Jeonbuk National University

Round 2

Reviewer 2 Report

The authors have made efforts to correct the errors I have found in the past. Errors made by other reviewers have also been corrected. Thus, the writing already meets the expectations of "Materials" and shows the level that can be expected from a review article. I consider the article to be suitable for display in its present form.

Reviewer 4 Report

The manuscript is revised according to the comments and can be accepted.